# Diode Laser Overtone Spectroscopy of Methyl Iodide at 850 nm

Alessandro Lucchesini 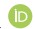

Istituto Nazionale di Ottica del Consiglio Nazionale delle Ricerche (INO–CNR), S.S. "Adriano Gozzin" Pisa, Via Moruzzi 1, 56124 Pisa, Italy; lucchesini@ino.cnr.it; Tel.: +39-050-621-2533

**Abstract:** Using Tunable Diode Laser Absorption Spectroscopy (TDLAS), 82 $CH_3I$ overtone absorption lines were detected for the first time in the range between 11,660 and 11,840 $cm^{-1}$ (844–857 nm), with strengths estimated around $10^{-27}$–$10^{-26}$ cm/molecule. The lines were measured utilizing commercial heterostructure F–P type diode lasers, multipass cells, and wavelength modulation spectroscopy with the second harmonic detection technique. A high modulation amplitude approach was adopted for the analysis of the line shapes. Self-broadening coefficients were obtained for two lines.

**Keywords:** methyl iodide; overtone band; high-resolution spectroscopy; tunable diode laser; line shape and width

## 1. Introduction

This work on methyl iodide or iodomethane ($CH_3I$) continues the previous work on methyl halides $CH_3F$ [1] and $CH_3Cl$ [2] ($C_{3v}$ symmetry group), detected at 850 nm in the gas phase by the author. In fact, their absorption bands in the infrared (IR) and in the near-infrared (NIR) part of the e.m. spectra are very similar.

Methyl iodide is a prolate symmetric top molecule, which is used in agriculture as a pesticide and is present in the earth's atmosphere, classified as a halogenated volatile organic compound (HVOC). It contributes to ozone layer depletion [3]. This molecule is one of the most studied symmetric tops in the IR, where it can be finely studied by spectroscopy using semiconductor sources.

The $CH_3I$ absorption spectrum between 850 nm and 2.5 μm has been detected previously by Gerhard and Luise Herzberg [4] using an infrared prism spectrometer, when they classified the observed overtone and combination bands. Even older is the work of Verleger on methyl halides at wavelengths below 1.2 μm [5], where overtone bands were observed on photographic plates through a 3 m grating monochromator. Methyl iodide optical absorption was studied more recently in the NIR by Ishibashi and Sasada [6] with diode lasers as the sources to detect the $2\nu_4$ overtone band at 6050 $cm^{-1}$ with sub-Doppler resolution in a Fabry–Perot (F.–P.) type cavity measurement cell.

In this experimental work, a tunable diode laser spectrometer (TDLS) with frequency modulation and the second harmonic detection technique was used to observe the $CH_3I$ ro-vibrational band at around 11,740 $cm^{-1}$ (850 nm), with a resolution of 0.01 $cm^{-1}$. This NIR band is presumably related to the third overtone of the $\nu_1$ quanta of C–H stretching excitation [4] or a combination of overtones, such as, for instance, $2\nu_1 + 2\nu_4$. This may imply overlapping bands, whose upper states are often coupled by various interactions, such as the Fermi and the Coriolis resonances [7]. For this reason it is difficult to identify the right quanta of vibration and rotation, unless sophisticated techniques such as microwave-optical double resonance [8] are used.

This band is very weak, as the dipole moment is even less than the one of $CH_3F$ and $CH_3Cl$, as the C–I bond length is shorter than the C–F and C–Cl ones; therefore, a lower absorption strength for this molecule is expected. It is therefore necessary to use long optical paths and noise reduction techniques to observe and study these absorption lines. Frequency modulation spectroscopy [9], conventionally called Wavelength Modulation

Spectroscopy (WMS) when the value of the frequency of the modulation is chosen much lower than the sampled line width, was used here as the noise reduction technique.

In the following, we see how this technique allowed detection for the first time of the very weak ro-vibrational $CH_3I$ lines at 850 nm and to study their behavior at different gas pressures.

## 2. Experimental Details

The adopted experimental setup is sketched in Figure 1 and follows the previous work on $CH_3Cl$ [2]. The employed source was the AlGaAs/GaAs double heterostructure F.–P. type diode laser (DL), Thorlabs L852P100, with single longitudinal and transverse emission mode and maximum power $\simeq$100 mW cw in *free-running* configuration.

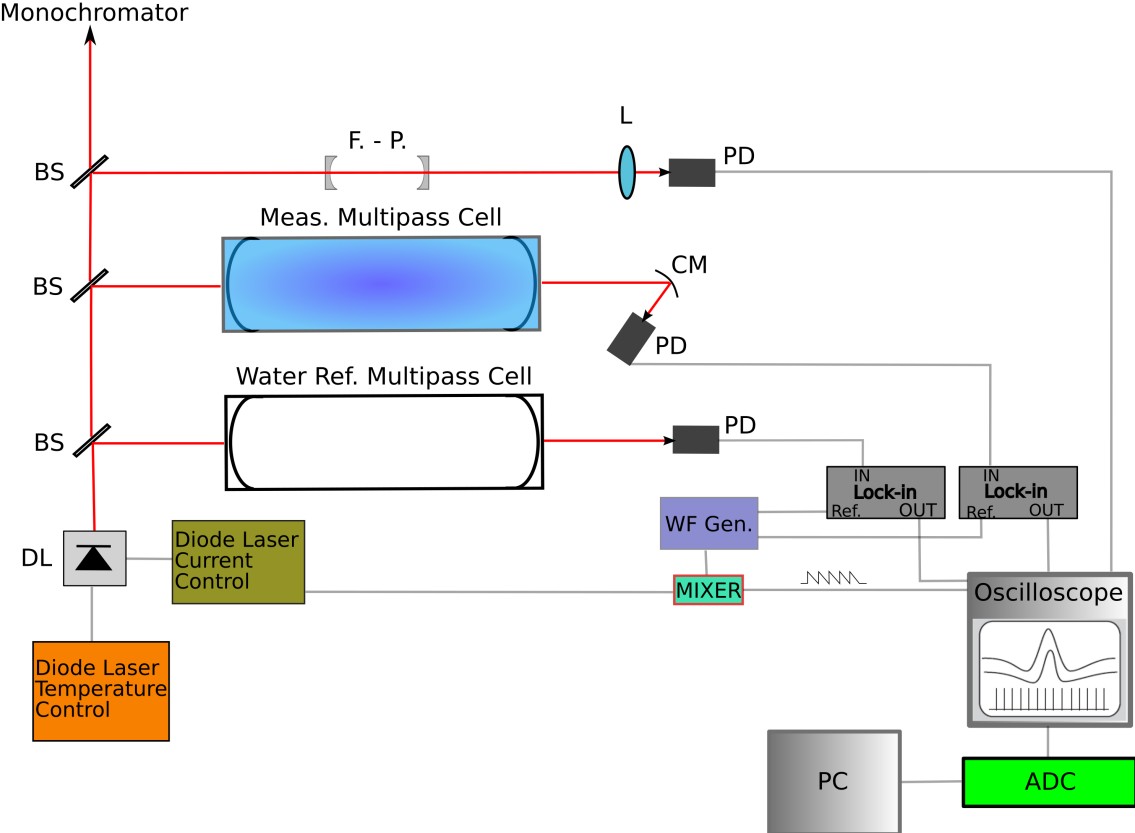

**Figure 1.** Outline of the experimental apparatus. ADC: analog-to-digital converter; BS: beam splitter; CM: concave mirror; DL: diode laser; F.– P.: confocal Fabry–Perot interferometer; L: lens; PC: desktop computer; PD: photodiode; WF Gen.: waveform generator.

The DL temperature control is very critical, as its typical emission wavelength varies 0.1 nm/K; therefore, a customized bipolar temperature controller was used to drive a Peltier junction coupled to the DL mount. This guaranteed a stability of ∼0.002 K per hour. Moreover, the DL current needs precise and fine control, since the characteristic slope of the DL emission is about 0.01 nm/mA. Consequently, a low noise DL current controller was needed: for this experiment, a custom-made precise current source was used, operating between 0 and 250 mA, with an accuracy of ±2.5 μA. To modulate and sweep the source frequency, a sine wave carrier signal from a low noise waveform generator was mixed with a ramp extracted from the oscilloscope sawtooth signal and fed to the DL current controller.

Two custom Herriott-type astigmatic multipass cells with optical path lengths of 30 m were used to house the sample and reference gases. The latter contained water

vapor at room temperature (RT) with a partial pressure $\simeq 20$ torr and was used as the reference gas for wavenumber measurements (for this purpose, the HITRAN2016 molecular spectroscopic database [10] was adopted) and to verify whether the water absorption lines could interfere with the observed $CH_3I$ lines. At the exit of the measurement cell, a concave mirror focused the laser beam on the active spot of the photodiode (PD), with the purpose of reducing the mechanical noise originating from the optical leverage of the spectroscope set.

The detectors chosen for the purpose were preamplified silicon PDs (Centronic OSD5-5T) with an active surface of 2.52 mm in diameter, whose outputs were sent to lock-in amplifiers tuned to the carrier frequency ($f \sim 5$ kHz) from the waveform generator. It was verified that the residual phase errors in the synchronization of the two lock-in amplifiers could induce at most a corresponding wavenumber error of 0.003 cm$^{-1}$. A confocal 5 cm F.–P. interferometer (f.s.r. = 0.05 cm$^{-1}$) was utilized to check the DL emission mode and the linearity of its emission frequency, while a 0.35 m focal length Czerny–Turner monochromator with 1180 lines per mm grating was used for the rough wavelength check ($\pm 0.01$ nm).

The vacuum in the sample cell was obtained by using a double stage rotary pump with limit pressure $<1 \times 10^{-4}$ torr, and the pressure inside the cell was measured directly by a capacitive pressure gauge (accuracy $\pm 0.5$ torr). All experiments were conducted at room temperature and at pressure values ranging from 20 to 90 torr. Analytical grade methyl iodide (Sigma-Aldrich, 99% purity, with silver as stabilizer) was used as supplied. The sample was contained in a stainless steel tube and allowed to flow into the evacuated measuring cell at its vapor pressure.

*Wavelength Modulation Spectroscopy*

Let us describe the transmittance $\tau(\nu)$ by the Lambert–Beer expression

$$\tau(\nu) = e^{-\sigma(\nu)z}, \tag{1}$$

where $z = \rho\,l$ is the product of the density of the absorbing species $\rho$ and the optical path $l$ of the radiation through the sample, and $\sigma(\nu)$ is the absorption cross section, which follows the behavior of the line shape. In regimes of small optical density [$\sigma(\nu)z \ll 1$], as in most experimental conditions, Equation (1) becomes

$$\tau(\nu) \simeq 1 - \sigma(\nu)\,z. \tag{2}$$

The WMS technique applied was achieved here by modulating the the source emission frequency $\bar{\nu}$ at $\nu_m = \omega_m/2\pi$, with amplitude $a$:

$$\nu = \bar{\nu} + a\cos\omega_m t. \tag{3}$$

The transmitted intensity could then be written as a cosine Fourier series:

$$\tau(\bar{\nu} + a\,\cos\omega_m t) = \sum_{n=0}^{\infty} H_n(\bar{\nu}, a)\cos n\omega_m t, \tag{4}$$

where $H_n(\bar{\nu})$ is the $n$-th harmonic component of the modulated signal. With a lock-in amplifier tuned to a multiple $n\nu_m$ ($n = 1, 2, ...$) of the modulation frequency, an output signal proportional to the $n$-th component $H_n(\bar{\nu})$ was obtained. In the case where the value of $a$ is chosen smaller than the line-width, the $n$-th Fourier component is proportional to the $n$-th derivative of the original signal:

$$H_n(\bar{\nu}, a) = \frac{2^{1-n}}{n!}\,a^n\,\left.\frac{d^n\tau(\nu)}{d\nu^n}\right|_{\nu=\bar{\nu}}, \qquad n \geq 1. \tag{5}$$

In order to detect very low absorbances, a high modulation amplitude regime is required, which implies a modulation index $m \equiv a/\Gamma \gg 0.1$, where $\Gamma$ is the absorption line width.

Line position measurements were carried out at pressures around 30 torr. In this condition, where the collisional effect was not negligible, the absorption line shape was well described by the Voigt profile, a convolution of the Gaussian (Doppler regime) and Lorentzian (collisional regime) functions:

$$f(v) = \int_{-\infty}^{+\infty} \frac{\exp\left[-(t - v_\circ)^2/\Gamma_G^2 \ln 2\right]}{(t - v)^2 + \Gamma_L^2} dt, \tag{6}$$

where $v_\circ$ is the gas resonance frequency, and $\Gamma_G$ and $\Gamma_L$ are the Gaussian and the Lorentzian half-width at half-maximums (HWHMs), respectively. Second-order effects, such as a velocity-changing collision or Dicke narrowing [11], were not observed within the sensitivity of the apparatus and therefore were not taken into account.

The phase detection technique, obtained by tuning the lock-in amplifiers to twice the modulation frequency ($\sim 10$ kHz), produced a line-shape signal that was closer to the second derivative of the absorption feature, as the minor was the amplitude of the modulation. The symmetry of the line shape was not perfect due to the simultaneous amplitude modulation of the source as the DL injection current varied; however, this effect becomes negligible in high modulation regimes. This "2$f$ detection" brings the advantage of a flat baseline of the signal but cannot avoid optical interferences, coming from the many reflecting surfaces present in the optical path. In this specific case, this drawback limited the detection sensitivity to absorbances higher than $1 \times 10^{-7}$. Under these conditions, this technique could not provide a reliable measurement of the intensity parameter when applied to very weak resonances. For this reason, it was possible only to estimate the strengths of the $CH_3I$ absorption lines, which here appeared to fall in the range $1 - 30 \times 10^{-27}$ cm/molecule from a comparison with the absorption line strengths of the water vapor present in the same spectrum, when a known partial pressure of this gas was added into the measurement cell. The intensity distribution of the observed $CH_3I$ lines showed a typical behavior of the ro-vibrational spectra, centered around 11,740 cm$^{-1}$, where the strongest lines were found.

To obtain line positions and widths with good reliability even for the weakest lines, the values of the modulation index $m$ were set around 2.0–2.3. This improved the S/N ratio but did not allow the use of Equation (5) any more. The approximate function that represented the absorption line distorted by the modulation was specifically evaluated and is reported in the Appendix A. A nonlinear least squares fit method was applied to extract the line parameters from the collected profiles.

Finally, the following expression of the collisional half-width at half-maximum (HWHM), as a function of pressure, was adopted for the line broadening calculations:

$$\Gamma_L(p) = \gamma_{\text{self}}\, p, \tag{7}$$

where $p$ is the sample gas pressure, and $\gamma_{\text{self}}$ is the gas self-broadening coefficient.

## 3. Experimental Results

The 82 detected $CH_3I$ absorption lines are listed in Table 1, where the maximum error on the wavenumber ($v'$) was within the second decimal unit, referring to the ten times more precise $H_2O$ atlas [10]. The wavelengths are reported for convenience in air at $T$ = 294 K, following the work of Edlén [12].

Figure 2 presents the 2$f$ absorption measurement signal for $CH_3I$ at 11,789.28 cm$^{-1}$, $T$ = 294 K, $p_{CH_3I} = 31$ torr, and $m \simeq 2$, along with the fit and its residuals. It also displays the $H_2O$ 2$f$ reference signal at 11,789.41 cm$^{-1}$ and the transmission of the F.–P. interferometer used for frequency linearization. The evident etalon effect in the $CH_3I$ measurement plot originated from the reflections within the optical path, and it was taken into account in the fit procedure. Only the higher frequency fringes remained in the fit residuals, originating from the confocal mirrors of the multipass cell; in fact, the distance between them was $l = 42.8$ cm and $\Delta v' \cong 1/(4l) = 0.0058$ cm$^{-1}$.

**Table 1.** Wavenumbers and wavelengths (in air at room temperature) of the detected $CH_3I$ absorption lines, with the maximum error within the second decimal unit.

| $\nu'$ (cm$^{-1}$) | $\lambda$ (Å) | $\nu'$ (cm$^{-1}$) | $\lambda$ (Å) | $\nu'$ (cm$^{-1}$) | $\lambda$ (Å) | $\nu'$ (cm$^{-1}$) | $\lambda$ (Å) |
|---|---|---|---|---|---|---|---|
| 11,659.92 | 8574.08 | 11697.76 | 8546.35 | 11,720.95 | 8529.44 | 11,772.70 | 8491.94 |
| 11,660.13 | 8573.93 | 11,698.02 | 8546.16 | 11,730.12 | 8522.77 | 11,777.29 | 8488.63 |
| 11,664.91 | 8570.41 | 11,700.55 | 8544.31 | 11,730.34 | 8522.61 | 11,777.94 | 8488.17 |
| 11,665.03 | 8570.33 | 11,700.85 | 8544.09 | 11,730.43 | 8522.54 | 11,778.05 | 8488.09 |
| 11,667.27 | 8568.68 | 11,700.95 | 8544.02 | 11,730.62 | 8522.41 | 11,778.27 | 8487.93 |
| 11,683.78 | 8556.57 | 11,704.88 | 8541.15 | 11,738.96 | 8516.35 | 11,785.46 | 8482.75 |
| 11,683.92 | 8556.47 | 11,706.39 | 8540.04 | 11,739.46 | 8515.99 | 11,785.65 | 8482.61 |
| 11,684.03 | 8556.39 | 11,706.48 | 8539.98 | 11,739.58 | 8515.90 | 11,786.01 | 8482.35 |
| 11,684.18 | 8556.28 | 11,706.55 | 8539.94 | 11,739.69 | 8515.82 | 11,788.60 | 8480.49 |
| 11,684.43 | 8556.10 | 11,706.77 | 8539.77 | 11,739.80 | 8515.74 | 11,789.28 | 8480.00 |
| 11,684.58 | 8555.99 | 11,709.79 | 8537.57 | 11,741.14 | 8514.77 | 11,798.00 | 8473.73 |
| 11,684.74 | 8555.87 | 11,709.94 | 8537.46 | 11,741.39 | 8514.59 | 11,805.97 | 8468.01 |
| 11,684.95 | 8555.41 | 11,710.29 | 8537.20 | 11,748.32 | 8509.57 | 11,806.13 | 8467.90 |
| 11,686.83 | 8554.34 | 11,710.55 | 8537.01 | 11,761.04 | 8500.36 | 11,826.25 | 8453.49 |
| 11,687.14 | 8554.11 | 11,711.17 | 8536.56 | 11,761.75 | 8499.85 | 11,827.97 | 8452.26 |
| 11,687.44 | 8553.89 | 11,711.41 | 8536.38 | 11,767.15 | 8495.95 | 11,841.21 | 8442.81 |
| 11,687.72 | 8553.69 | 11,711.53 | 8536.30 | 11,767.29 | 8495.85 | 11,841.39 | 8442.68 |
| 11,694.13 | 8549.00 | 11,711.64 | 8536.22 | 11,767.42 | 8495.75 | 11,841.46 | 8442.63 |
| 11,694.39 | 8548.81 | 11,713.72 | 8534.70 | 11,768.03 | 8495.31 | 11,842.02 | 8442.23 |
| 11,694.67 | 8548.61 | 11,713.81 | 8534.64 | 11,768.13 | 8495.24 | | |
| 11,697.42 | 8546.59 | 11,720.67 | 8529.65 | 11,770.45 | 8493.57 | | |

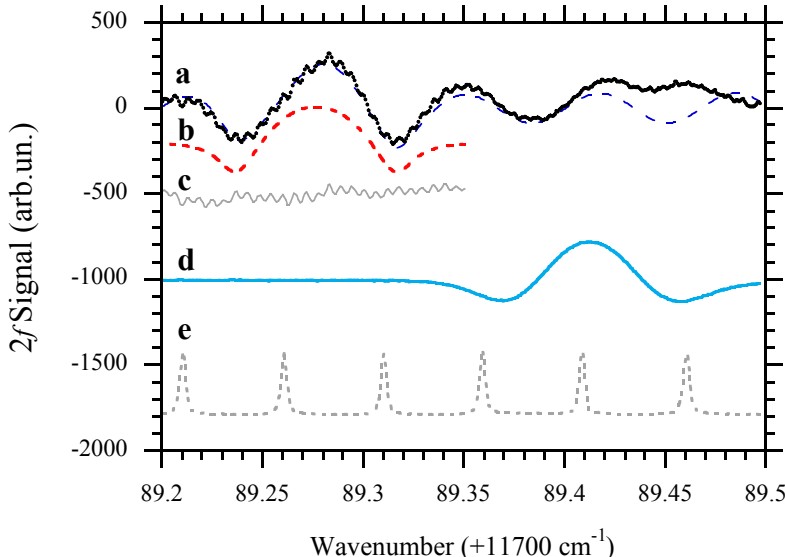

**Figure 2.** Second harmonic absorption signal of $CH_3I$ around 848 nm (black dots) at $p = 31$ torr and RT, with the best fit (dashed blue line) (**a**), the extracted peak fit (**b**), and the residuals (**c**), along with the $H_2O$ reference signal (**d**), all obtained by WMS with 10 Hz bandwidth. The F.–P. interferometer transmission (f.s.r.= 0.05 cm$^{-1}$) is also shown (**e**).

The self-broadening coefficients ($\gamma_{self}$) were measured for the first time for two $CH_3I$ absorption lines at RT and are shown in Table 2.

**Table 2.** Measured $CH_3I$ self-broadening coefficients (HWHM).

| $\nu'$ (cm$^{-1}$) | $\gamma_{self}$ (cm$^{-1}$/atm) |
|---|---|
| 11,741.39 | $0.23 \pm 0.02$ |
| 11,778.27 | $0.18 \pm 0.02$ |

The self-broadening measurement for the line at 11,741.39 cm$^{-1}$ is shown in Figure 3, where the Lorentz component of the absorption line width is plotted as a function of methyl iodide pressure at RT.

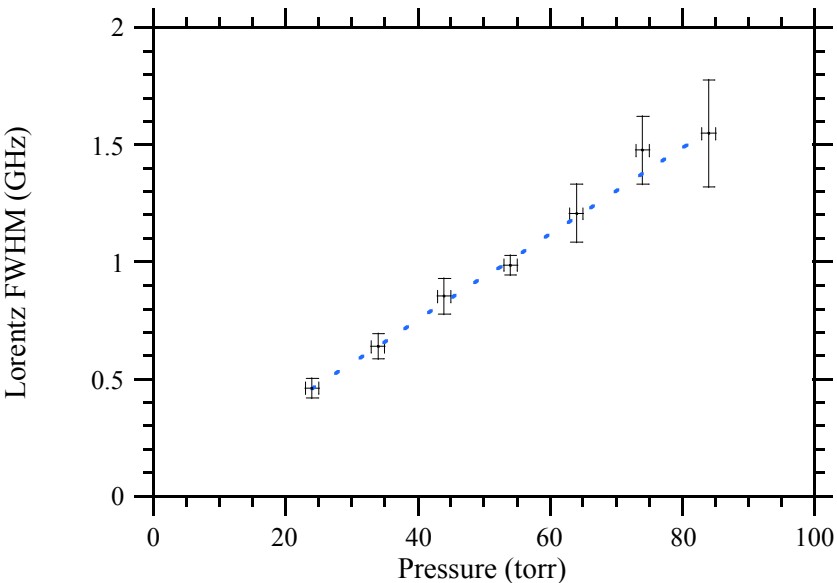

**Figure 3.** Self-broadening measurements on the methyl iodide 11,741.39 cm$^{-1}$ absorption line at RT obtained by TDLS and the procedure explained in the text. The blue dashed line shows the best linear fit.

In the literature, we did not find any measurements of pressure line-broadening at these wavenumbers, but a comparison can be attempted with what was obtained in other spectral regions.

From the HITRAN database [10], for the $\nu_4$ fundamental band at 3.3 μm, the $\gamma_{\text{self}}$ is reported between 0.1 and 0.5 cm$^{-1}$/atm at 296 K. Hoffman et al. [13], using diode laser absorption spectroscopy, observed $\gamma_{\text{self}}$ at RT for the $\nu_5$ band at 7 μm from 0.10 to 0.45 cm$^{-1}$/atm. Raddaoui et al. [14], using a Fourier transform spectrometer at RT, obtained $\gamma_{\text{self}}$ ranging from 0.10 cm$^{-1}$/atm (at low and high quantum rotational parameter *J*) to 0.45 cm$^{-1}$/atm for the $\nu_6$ fundamental roto-vibrational band of CH$_3$I at around 11 μm. Again, in this ro-vibrational band at RT, Attafi et al. [15] obtained $\gamma_{\text{self}}$ from 0.15 to 0.36 cm$^{-1}$/atm. Belli et al. [16], using the Doppler-free double-resonance technique, measured the self-collisional broadening parameters at RT for some absorption lines in the $\nu_6$ fundamental band, presenting values ranging from 0.18 to 0.20 cm$^{-1}$/atm. Finally, Ben Fathallah et al., using Fourier transform spectroscopy at RT, obtained self-broadening coefficient values from 0.14 to 0.36 cm$^{-1}$/atm for the bands $\nu_5$ and $\nu_3 + \nu_6$ [17]; while for the band $\nu_2$ [18] at around 8 μm, they obtained an average value of $\gamma_{\text{self}}$ equal to 0.25 cm$^{-1}$/atm, with a minimum at 0.1 cm$^{-1}$/atm and a maximum at 0.4 cm$^{-1}$/atm.

Our results were all within these ranges.

### 4. Conclusions

In total, 82 new CH$_3$I absorption lines between 11,660 and 11,840 cm$^{-1}$ were measured for the first time with a precision of 0.01 cm$^{-1}$, using a tunable diode laser spectrometer, wavelength modulation spectroscopy, and the second harmonic detection technique in a 30 m Herriott-type multipass cell. A high modulation amplitude approach was adopted, and a dedicated fit function was developed. The strength of the observed lines varied between 10$^{-27}$ and 10$^{-26}$ cm/molecule at room temperature. For two of the observed lines, self-broadening coefficients similar to those reported for the same molecule in other spectral regions were obtained for the first time.

**Funding:** This research received no external funding.

**Institutional Review Board Statement:** Not applicable.

**Informed Consent Statement:** Not applicable.

**Data Availability Statement:** The data are contained within the article.

**Acknowledgments:** The author wishes to thank A. Barbini for the electronic advising, as well as M. Tagliaferri and M. Voliani for the technical assistance.

**Conflicts of Interest:** The author declares no conflicts of interest.

## Abbreviations

The following abbreviations are used in this manuscript:

| | |
|---|---|
| MDPI | Multidisciplinary Digital Publishing Institute |
| DL | Diode laser |
| F–P | Fabry–Perot |
| HVOC | Halogenated volatile organic compound |
| HWHM | Half-width at half-maximum |
| IR | Infrared |
| NIR | Near-infrared |
| PD | Photodiode |
| RT | Room temperature |
| S/N | Signal-to-noise |
| TDLAS | Tunable diode laser absorption spectroscopy |
| WMS | Wavelength modulation spectroscopy |

## Appendix A. Frequency Modulation in the High-Amplitude Regime

The use of high modulation amplitude $a$ is a necessity in order to increase the S/N ratio for very weak absorption lines. In this case the derivative approximation of Equation (5) no longer works, and it is more appropriate to start from the other expression [19]:

$$H_n(\nu, a) = \frac{2}{\pi} \int_0^\pi \tau(\nu + a \cos \theta) \cos n\theta \, d\theta.$$ (A1)

In order to extract the collisional component from the absorption line-shape, Arndt [20] and Wahlquist [21] derived the analytic form of the harmonics for a Lorentzian function, which is the right choice when dealing with collisional broadening.

To do this, they obtained the $n$th harmonic element inverting Equation (4):

$$H_n(x, m) = \varepsilon_n \, \mathrm{i}^n \int_{-\infty}^{+\infty} \hat{\tau}(\omega) \, J_n(m\omega) \, e^{\mathrm{i}\omega x} \, d\omega,$$ (A2)

where

$$\hat{\tau}(\omega) = \frac{1}{2\pi} \int \tau(x) \, e^{-\mathrm{i}\omega x} \, dx$$ (A3)

is the Fourier transform of the transmittance profile; $x = \nu/\Gamma$ and $m = a/\Gamma$, $\Gamma$ is the line width; $J_n$ is the $n$th order Bessel function; $\varepsilon_0 = 1$, $\varepsilon_n = 2$ ($n = 1, 2, \cdots$); and i is the imaginary unit. The absorption cross section in Equation (2) is then put into the Lorentzian form:

$$\sigma_{\mathrm{L}}(x, m) \propto \frac{1}{1 + (x + m \cos\omega t)^2}.$$ (A4)

The second Fourier component of the absorption cross section can be recalculated following Arndt's work by setting $n = 2$:

$$H_2(x, m) = -\frac{1}{m^2} \left[ \frac{\{[(1 - \mathrm{i}x)^2 + m^2]^{1/2} - (1 - \mathrm{i}x)\}^2}{[(1 - \mathrm{i}x)^2 + m^2]^{1/2}} + \mathrm{c.c.} \right]$$ (A5)

and by eliminating the imaginary part:

$$H_2(x,m) = \frac{2}{m^2} - \frac{2^{1/2}}{m^2} \times$$
$$\frac{1/2[(M^2+4x^2)^{1/2}+1-x^2][(M^2+4x^2)^{1/2}+M]^{1/2}+|x|\,[(M^2+4x^2)^{1/2}-M]^{1/2}}{(M^2+4x^2)^{1/2}},$$

(A6)

where

$$M = 1 - x^2 + m^2.$$

Equation (A6), close to the second derivative of the absorption feature only for low $m$, simulates the behavior of the line shape at high modulation amplitudes. This is shown in Figure A1, where the equation is plotted in 3D, varying the modulation index $m$. For $m = 3$, the second derivative is completely deformed by broadening, as occurs in reality.

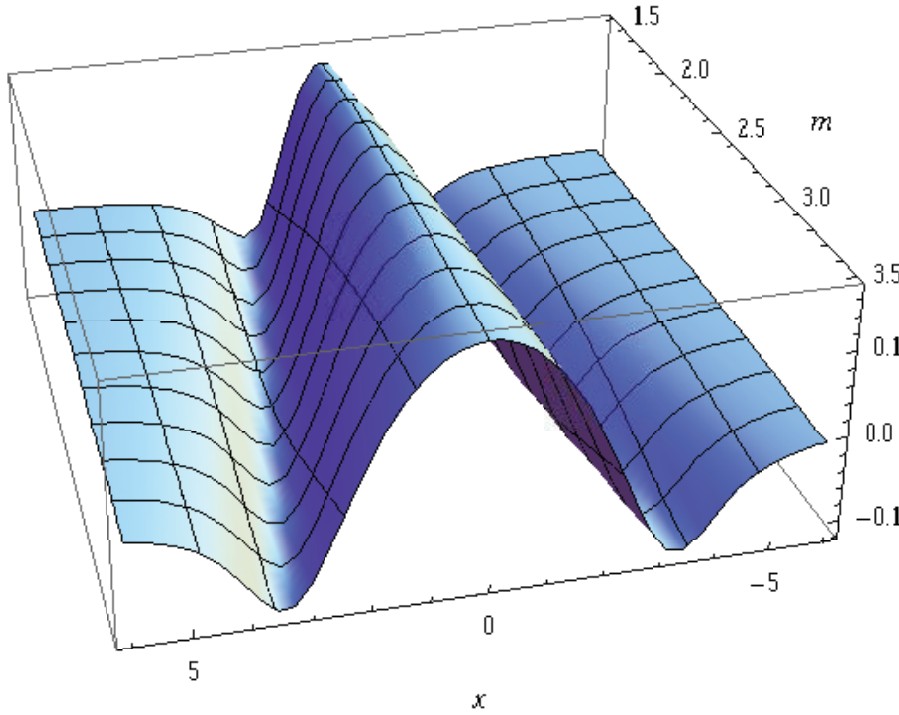

**Figure A1.** Behavior of Equation (A6) as a function of the modulation parameter $m$.

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
