# Peer review of "Diode Laser Overtone Spectroscopy of Methyl Iodide at 850 nm"

_2813-446X, doi:10.3390/spectroscj1010003_

Round 1
Reviewer 1 Report
Report on the manuscript titled "Diode Laser Overtone Spectroscopy of Methyl Iodide at 850 nm" by Alessandro Lucchesini.
The author has measured 82 new lines of the IR spectrum of methyliodide. These lines arise from overtone transitions and are extremely weak. Furthermore, the spectrum is perturbed due to Fermi resonances, for this reason the lines may perhaps not be attributable to transitions between vibrational and rotational states in terms of the usual spectroscopic quantum numbers.
I note that no intensities are reported in Table 1. Could the author provide such information or comment on this?
Please note that the first name of Prof. G. Herzberg was Gerhard.
I recommend publication of this manuscript, as these lines have never been
measured before and their positions are accurate to 0.01 cm-1, hopefully
with intensities added to Table 1.
Author Response
I changed Gerard in Gerhard.
I decided not to directly publish the line-strength values of the single line in Table 1, because the procedure of comparison with the reference water vapor strengths in the same range gave errors greater than 50%.
Therefore I thought it more appropriate to give only an indicative estimate in the text.
At page 4, line 123, I added more information:
"The intensity distribution of the observed CH3I lines showed a typical behavior of the ro-vibrational spectra, centered around 11740 cm-1, where the strongest lines were found."
Reviewer 2 Report
The Manuscript titled “Diode Laser Overtone Spectroscopy of Methyl Iodide at 850 nm” submitted by Alessandro Lucchesini is an extension of his earlier published on Overtone spectroscopy of Methyl Fluoride and Methyl Chloride.
The experimental design is sophisticated and data collection for the overtone of methyl iodide is meticulous. However, there are a few questions, which I would like the author to address. I recommend the paper be published after the author has addressed the queries mentioned below.
1) On page 4, line 138, it is mentioned that “The maximum wavenumber (ν ′ ) error is 0.01 cm−1; ”. Thus, the error in the measurement is at the 2nd decimal place. However, in the data shown in table 1, all the wavenumbers are listed up to two decimal places. I want the author to clarify that if the error is in the 2nd decimal place, all the data shown in Table 1 should be up to only 1 decimal place. For eg. 11659.9 and not 11659.92.
2) Self-broadening coefficients for wavenumber 11741.39 (cm−1) is 0.23 (cm−1/atm), as shown in Table 2. The experiments were done between 20 to 90 torr. This implies that the Self-broadening coefficients for the above line at 90 torr will be 0.027 cm−1. But as mentioned in comment 1, the maximum error is 0.01 cm−1. I would like the author to clarify the method for maximum error calculation, and if it takes into account the Self-broadening coefficients.
3)I would like the author to add relevant references for “Methyl iodide is used in agriculture as a pesticide and is present in the earth’s atmosphere classified as halogenated volatile organic compound (HVOC). It participates to the ozone layer depletion.”, as mentioned in the Introduction section.
Author Response
Answer:
1)
I substituted:
"The maximum wavenumber (n') error is 0.01 cm-1"
with
"the maximum error on the wavenumber (n') is within the second decimal unit"
Also in Table I.
2)
The pressure broadening measurements data for the 11741.39 cm-1 line are:
-------------------------------------------------------------------------
P ΓL(FWHM) Error(1 Sigma) Fit
(Torr) (GHz) (GHz) (GHz)
------------------------------------------------
24 0.46248 0.04182 0.45892
34 0.63960 0.05412 0.64222
44 0.85362 0.07626 0.82552
54 0.98728 0.04182 1.00882
64 1.20786 0.12300 1.19212
74 1.47682 0.14514 1.37542
84 1.54898 0.22878 1.55872
------------------------------------------------
As stated in the text at page 3, line 102, the line position measurements were carried on at pressures around 30 torr.
This is a good compromise to obtain the biggest S/N ratio.
As seen in Figure 3, increasing the pressure, the error increase as a consequence of the derivative based detection technique.
In this case, at 34 torr the error (FWHM) was = 0.05412 GHz, which corresponds to
0.05412 / 30 / 2 = 0.0009 cm-1 (1 Sigma) HWHM
3 Sigma: 0.0027 cm-1 (Max error)
For scruple, now in the test it is written that the maximum error is within the second decimal unit for all the lines.
3)
Added a new reference useful for the second paragraph of the Introduction:
"It participates to the ozone layer depletion [Ref.3]."
Author Response
Added a useful reference in the second paragraph of the Introduction:
"It participates to the ozone layer depletion [Ref.3]."
Then I made the corrections suggested in "spectroscj-2305204-review.pdf", some of which can be checked below.
In particular:
1.
At the end of the Introduction I added:
"In the following we will see how this technique allowed to detect for the first time very weak ro-vibrational CH3I lines at 850 nm and to study their behavior at different gas pressures."
13.
At page 6, I changed the expression "with what obtained", modifying the paragraph in:
"In the literature we have not found any measurements of pressure line-broadening at these wavenumbers, but a comparison can be attempted with what has been obtained in other spectral regions."
20.
In the conclusions I changed:
"A high modulation approach"
in
"A high modulation amplitude approach"
21.
"For two of them"
in
"For two of the observed lines".